# Determinants and Values of Willingness to Pay for Water Quality Improvement: Insights from Chia Lagoon, Malawi

**Rodgers Makwinja** [1,2], **Ishmael Bobby Mphangwe Kosamu** [2] and **Chikumbusko Chiziwa Kaonga** [2,*]

1   African Center of Excellence for Water Management, Addis Ababa University,
    P.O. Box 1176 Addis Ababa, Ethiopia
2   Department of Physics and Biochemical Sciences, University of Malawi, The Polytechnic,
    Private Bag 303, Chichiri, Blantyre 3, Malawi
*   Correspondence: ckaonga@poly.ac.mw

**Abstract:** Water resources in the Chia lagoon in Malawi experience a possible threat to sustainability. Communities are seeking alternatives to improve water quality in the lagoon. This study quantified the communities' willingness-to-pay (WTP) and their influencing factors while using contingent valuation (CV) techniques. A wide range of data collection procedures, including focus group discussions, key informant interviews, field observation, and CV survey, were employed. A sample of 300 households was randomly selected. The CV results showed that 57.4% of the households were willing to pay. The monthly individual aggregate WTP amount ranged from MK696.83 (US$0.95) to MK81697 (US$111.38), and on average MK7870.45 (US$10.73), generating aggregate annual values ranging from MK6, 689,568 (US$9126.29) to MK784, 294,080 (US$1,069,978), and on average MK75,556,320 (US$103,078) (ceteris paribus). Logistic regression model demonstrated a significant ($p < 0.01$ or $p < 0.05$) relationship between demographic (gender, age, literacy level), social-economic (land ownership, main agriculture water source, and income), and institutional (civic education and social network, extension, institutional trust, household socio trust) factors and WTP. The findings from this study provide significant clues for further research and baseline information for local government and communities in the development of more effective and holistic approaches for improving water quality in natural ecosystems.

**Keywords:** contingent valuation; water resources; water quality; willingness-to-pay (WTP); logistic regression

## 1. Introduction

Water resources in the natural ecosystems are considered as vital socio-economic and environmental goods at the global scale [1,2]. As reported by Wasambo [3], water has a capacity to stimulate economic development, particularly in developing countries. It is further noted that prospects of poverty reduction and economic growth are highly dependent on water resources [4]. The countries whose major economic activity is agriculture are mostly susceptible to the unpredictability of water availability both in time and space [5]. Currently, water scarcity has become a critical issue in many developing countries in Africa [6]. Tremendous increase in water demand, accompanied by rapid population expansion as well as economic explosions, has led into the collapse of the resource. Globally, it is estimated that about 3.6 billion people live in areas that face water scarcity for at least one month each year and this number is projected to increase to around 4.8–5.7 billion by 2050 [6]. On the other hand, Ward [7] projected that, by 2050, about 40% of the world's population is expected to experience serious water scarcity.

Malawi is amongst the countries in the world endowed with a variety of fresh water systems [8]. The water systems in Malawi include Lake Malawi, Lake Malombe, and Shire River, which forms part of African great rift valleys, Lake Chilwa, Lake Chiuta, Chia lagoon, and several networks of river systems [9]. Although, Malawi is known to have variety of water systems, it is documented as a water-stressed country, and is projected to be water scarce by 2025 [10,11]. In Malawi, rapid population growth [12], climatic conditions [13], environmental degradation [14], rural poverty, unsustainable use of water resources in the natural ecosystems, poor forestry, and agricultural practices [15–17] have led to the depletion and degradation of the resource. The consequences have been frequent disease outbreak, repeated famine [18], food crises, loss of biodiversity [14,19,20], water scarcity, and conflicts over water use [14]. Water resources at Chia lagoon in Central Malawi provide an excellent illustration of a crisis in water resources management. With no clearly well-defined strategy and management plan to protect the resources, the lagoon has been overstressed from pollution, salinization, and weed invasion [19]. The impact of climate change, such as upward trend of extreme weather-related events, has further stressed the resources in the lagoon [10,19,21,22].

Given the tension between limited water endowment in the natural ecosystems and the increasing demand that is driven by human population growth and socio-economic development, the government of Malawi through various legal frameworks has taken measures to safeguard and protect water resources in the country, including the enactment of the Environment Management Act of 2017 [23]; the Forestry Policy of 1997 [24]; the Land Policy of 2002 [25]; the Water and Sanitation Policy of 2006 [26]; and, the National Water Policy (2005) [10]. However, most of these government legal frameworks attach the central role of managing the resource to the state overlooking the significance of stimulating public participation in the management of the resources [27]. Historically, restoration efforts of rivers, lagoons, lakes, wetlands, and other natural ecosystems in Malawi have been often embedded in hullabaloo emanating from budget constraints and disagreement over the gravity of the problems and appropriate policies. Although a significant number of studies using CVM to assess water quality improvement in coastal waters and estuaries have been recently conducted in Asia, Europe, America, and some parts of Africa [28–35], such studies are scarcely available in Malawi. Besides, useful information to be integrated in decision-support system to address changes in water quality in the natural ecosystems in Malawi barely exist.

Recently, environmental economists have conventionally managed to address water quality changes in the natural ecosystems by adopting approaches that rely upon WTP survey responses [28,36,37]. According to Martínez-Paza et al. [34], the WTP survey has been proven to be indispensable tool for decision–making process. It can promote public participation throughout the process, especially in African countries where additional constraints for improvement of rivers, lakes, and wetlands exist [34]. The willingness-to-pay (WTP) survey is further required in countries, such as Malawi, where there is increase in competition of water uses, severe water scarcity and high spatial and temporal variability, and social and political resistance to allocate flows for environmental aims, while restricting water use for agriculture or other use is very strong and public and stakeholder acceptance of environmental flows is much more difficult [36]. In this paper, the WTP for communities around Chia lagoon was estimated. The contingent valuation (CV) technique was used to provide necessary data. Demographic, socio-economic, and institutional variables and influencing factors that expound the willingness of individual household to pay for improved water quality in the lagoon were explored and identified. The aim of this paper was to contribute to an understanding of how the communities can embrace an effective water governance system in the natural ecosystems in order to achieve sustainable water resources management.

The subsequent sections of this paper are as follows: Section 2 provides theoretical framework, Section 3 describes research design and approaches, Section 4 illustrates study results, Section 5 illustrates and deduces the practical discussion, and Section 6 provide a brief summary of the research and policy suggestions.

## 2. Theoretical Framework

### 2.1. Hypothesis Underpinnings Household's WTP

Although the Malawi government developed Integrated water resources management and water efficiency plan (IWRM/WEP) with the aim of ensuring the coordinated development and management of water, land, and other related resources by maximizing economic and social-welfare without compromising the sustainability of environmental systems [35], policy makers continue to face challenges in setting national priorities and implementing effective and efficient water quality management programs. Evaluating the social and economic implications of water resources policy options has been proven difficult. Effort by the Malawi government to synthesize and integrate various natural, physical, and social sciences [35] in managing water resources in the natural ecosystems has been proven futile. Research has shown that, as population grows and human activities increase, water pollution and water demand for agriculture, domestic, industry, and other uses increases [35]. The valuation of these benefits is necessary to justify a suitable investment policy since water in the natural ecosystem is linked to many social and economic benefits.

Hernández-Sancho et al. [38] highlighted that the effective implementation of IWRM/WEP to prevent water resources degradation and depletion requires determining their value in social and economic terms and incorporating this information into the decision-making process. In other words, the task of satisfying an increasing demand for water resources while avoiding the degradation of the ecosystem requires viable answers from economic and environmental view point [36]. In Malawi, a limited number of studies exist regarding the subject of economic valuation [39]. In this paper, a shadow price was used to estimate WTP to improve water quality in Chia lagoon. The contingent valuation method was adopted to measure individuals' WTP [40]. According to Hynes et al. [41], CVM takes a more holistic approach by focusing on the value of moving from the status quo to an alternative status of the goods and services. The study design was based on the theory of community perception [42]. In Malawi, Limuwa et al. [22] adopted the same approach to evaluate small scale fishers' perception on climate change and their coping strategies. The willingness of communities to pay for improved water resources management depends on their perception towards the resource [42]. The philosophy of WTP originated from economic theory (consumer theory) [43]. It reflects the maximum amount of money that an individual is willing to give up to obtain more of the good or services [15]. The measure of economic value is expressed as WTP amount because money is used as a standard measure in this context. Contingent valuation survey (CVS) is used to determine the WTP amount.

### 2.2. Advantages and Disadvantages of CV Techniques

The economic valuation of water quality changes in the natural ecosystems relies on CV techniques. Contingent valuation technique is a social survey method where the individuals are presented with information regarding specific water quality changes, the value which cannot be accounted for in real economic markets [43]. In CV surveys, individual perception, attitude, and preferences regarding some water quality changes and its non-market values are elicited [44]. Hypothetical market place [45], in which no actual transactions are made [46], is created. The CV method asks people to report their WTP to have improved water quality in the lagoon. The technique emphasizes the stated preferences of respondents and it stands in contrast to those approaches that use revealed preferences [47]. Researchers have argued that CV method gives rise to implausible responses [48,49]. Others have also argued that the CV method is the perfect method for estimating private goods not public [50]. For public goods, it is argued that the actual WTP may understate compensating variation due to the free rider problem [51]. Aadland and Caplan [52] also argued that CV method is subjected to hypothetical bias. In other words, the method is subjected to any deviation of an individual's stated WTP from an actual WTP. Aadland and Caplan [52] further explained that hypothetical bias may arise from household's incentives to influence policy decisions. Attempts may be made to reduce the hypothetical bias by linking the survey

more closely to policy decision; however, this may also lead into strategic bias [53]. Despite these criticisms, it was noted that CV method had many advantages and a long history of use in major policy areas [47]. Chatterjee et al. [47] disputed the hypothetical bias by explaining that people do not always exaggerate their WTP in CV survey, simply because they do not pay. Schultz et al. [34] explained that the CV technique is frequently applied in non-market valuation of environmental assets and is widely used in the case of ecological restoration. On the other hand, Perman et al. [54] highlighted that elements typically described in the CV scenario are the environmental problems, such as impaired water quality, change in provision of public goods, and environmental policy. Mitchell and Carson [55] claimed that CVM provides a more accurate assessment of an individual's opinions. In other words, CVM provides a wide spectrum of both applied and methodological case studies dealing with a huge variety of different public assets and natural resources [56]. In estimating WTP, different CVM designs are used [51]. Loomis [56] indicated that open-ended (OE) and dichotomous choice (CD) are the most common designs. However, Loomis [56] recommended OE designs and further explained that OE designs outperform DC on temporal stability grounds. Brown et al. [57], on the other hand, found that DC designs resulted in a greater hypothetical error. In other words, the design had a greater difference between stated and actual WTP. OE design also offers a degree of flexibility in estimation of WTP amount curve [51]. However, in this study, DC was preferred over OE. Khan et al. [43] highly recommended DC due to the fact that it reduces the size of referendum-based welfare measures and at the same time improves their statistical efficiency. The technique produces more precise preferences and WTP values [43]. The open-ended elicitation format has been further criticized due to the fact that it produces larger numbers of non-response, zero, or protest responses as compared to DC, and it has been found to experience more uncertainty in answering OE than DC WTP questions [43].

### 2.3. Empirical Examples of Application of CV Techniques

Contingent valuation method is widely adopted in many environmental valuation studies. For instance, Alvarez-Farizo et al. [51] noted that CVM was one of the most common valuation methods used in Europe and the United States of America (USA). Bennett [52] used the CV techniques to evaluate the existence of preserved nature in Australia and was very supportive of the technique. Policy makers have also employed CV methods for assessing the demand of improved sanitation and water supply [46]. Furthermore, the CV method has been used to value reductions in acute and chronic illness and reduction in premature mortality associated with air quality improvements, [51]. The technique has again been used to assess the natural resources damages [58,59] and unit values of recreation [60]. Using the CV approach, Rodríguez-Tapia et al. [61] estimated household's Willingness to Pay for Clean Water in Mexico City. Chatterjee et al. [45] also used the CV method to estimate WTP for safe drinking water in Jacksonville, FL. Bilgic [62] further used the CV method to measure WTP to improve municipal water in southeast Anatolia, Turkey. Birdir et al. [63] used CVM to estimate WTP for coastal tourism management in Mersin, Turkey. On the other hand, Van Houtven et al. [29] estimated value of improving water quality in United States. Buckley et al. [40] used CVM to estimate WTP to achieve good status across Rivers in the Republic of Ireland. Jalilov et al. [64] used CVM to estimate WTP for water quality improvement in Metro Manila, Philippines. In this paper, the CV method was used in a survey to estimate WTP to improve water quality in Chia lagoon. It was assumed that WTP for improved water quality in the lagoon can help to identify and design policies for cost-effectiveness and sustainability of public resource [65].

### 2.4. Factors Explaining WTP for Water Quality Improvement

The second objective of this study was to theorize the potential factors affecting communities' WTP. Coastal areas, such as lagoons, are social-ecological systems that include both human and ecosystem components [66]. Therefore, understanding and managing natural ecosystems requires an integrative approach, which includes multiple disciplines, such as economics, sociology, and ecology [67]. In a public resource, such as water, the WTP of a household towards conservation of the

resources depend on several factors, amongst which being cultural, social, demographic, economic, and psychological factors [68]. Socio-economic variables are defined as factors that are associated with personal status within the community. de Groot et al. [68] argued that, in order to value ecosystems, we must first consider the ecological, socio-cultural, and economic values, and then assess the overall value as a reference for environmental decision-making management. Chapin, et al. [69] suggested that ecosystem services and biological diversity are intermediaries between the economic environment and human systems. It is presumed that a critical understanding of socio-economic factors can help to formulate appropriate policy that respond to the needs of the public.

Literature has further demonstrated that the rapid population growth in Malawi has environmental setbacks due to expansion of agriculture, settlements, and other livelihood activities [70]. Banderson, et al. [19] noted that rapid population growth and opening new land for farming, cultivation on steep slopes and stream bank, poor agriculture practices, felling of trees for wood, and setting bush fires that destroy vegetative cover have significantly contributed to the serious problems of water quality degradation in Chia lagoon. The consequences have been sedimentation of the lagoon, which has negatively affected its rich biodiversity and the livelihood of the local communities [19]. Communities have knowledge regarding how to improve water quality in the lagoon in case of unprecedented crises, such as increasing population pressure and famine [17,70]. Understanding key demographic factors influencing communities' decisions can help to improve water governance processes since population size, structure, and distribution have strong linkages with social and economic development, which in turn affect the management of natural resources, such as water in the natural ecosystems.

Again, institutional factors play a significant role in policy formulation and planning. Institutional factors, such as social networks, tie people together and stimulate mutuality [71]. Cook [72] accentuated the significance of institutional factors, such as socio-networks in solving local problems. In any community, common pool resources are controlled by local governance structures through land ownership or customary tenancy [73]. Though there are no private property rights, water resources, and other natural resources in Chia lagoon, socio-ecological systems are de facto properties for some households, chiefs, and governance structures. Kambewa [74] had similar observation in Lake Chilwa in Southern Malawi. Chiefs control land allocations in the lagoon periphery, while access to fish and other natural resources in the lagoon are controlled by local governance structures that are formed under co-management approach. Literature has demonstrated that over the last decade, the notions, policies, and practices of conservation of natural resources in many African countries, such as Malawi, have moved towards what has been regarded as community-based approach [75]. Critically, this may also be perceived as moving towards putting the community in conservation also termed as 'community-based conservation' [76] or participatory approaches, which is more complex than the dynamic of shifting of responsibility and authority from the state to the community [77]. Chia lagoon has various local governance structures, such as Beach village committees (BVC), Village Natural Resources Management Committees (VNRMC), Community Policing Committees (CPC), Fish Conservation Committees (FCC), government departments, and NGOs. Understanding key institutional factors influencing communities' decisions can help to improve water governance processes by facilitating the mobilization and allocation of key resources, such as water and conflict resolution, especially in managing the public resources, since Chia lagoon hosts a diverse local governance structure and various organizations operating at multiple scale under co-management approaches.

This study used an advanced econometric technique that lead to the development and application of Bernoulli regression model, 'a group of statistical models emanating from conditional Bernoulli distributions. Thus, logistic regression model to analyze the factors affecting WTP was applied. According to Greene [77], the logistic regression model is considered to be a perfect model when dealing with a categorical value. The model conforms to principles of homoscedasticity [78]. The model is further mostly preferred due to the fact that it is well fitted and suitable for describing and testing hypotheses regarding the relationship between dichotomous categorical outcome variables and one or more categorical predictor [79].

## 3. Research Design and Approaches

### 3.1. Study Area

The study was conducted at Chia lagoon, which lies between latitudes 13°0′ and 13°30′S, and longitudes 33°50′ and 42°20′E of Western Part of Lake Malawi. The area has an estimated population of 404,102, which is 2.3% of approximately 17 million Malawi population and receives average annual rainfall ranges from 860 to 1600 mm between December and March and average temperatures of 32 °C. Chia lagoon is the largest lagoon in Malawi with an approximately watershed area of 989 km$^2$ [15]. The lagoon is fed by the Lifuliza, Likoa, and Bambara Rivers, which originate from the Ntchisi hills through deeply incised gorges and valleys before winding through the lowland plains and entering the lagoon [19]. The Chia lagoon watershed has vast natural resources vital to over 55,000 human inhabitants [19]. The uplands are characterized by *Brachystegia julbernadia* savannah and woodland. The lagoon's fringes are heavily colonized by marsh reeds (*Phragmites spp.*) and shrubs that thrive under waterlogged conditions (e.g., *Aeschynomene, Mimosa* and *Sesbania spp*) [19].

### 3.2. Data Collection Protocol

The study applied a cross sectional sampling design. The data was collected on multiple occasions for an extended period of one year (January 2015 to December 2016). A combination of data collection procedures, such as reconnaissance surveys, focus group discussions, key informant interviews, field observation, and CV surveys were employed.

### 3.3. Reconnaissance Surveys

Six reconnaissance surveys in four villages within the study area were conducted from October 2015 to November 2015. Men and women were interviewed separately in order to capture the general perception of communities towards water situation in the lagoon. The interview was deliberated by asking the households to describe the changes of water quality at the lagoon for the past decades. The responses that were gathered from the exploratory surveys helped to identify critical issues and also to frame both qualitative and quantitative survey questionnaires [23]. The survey was carried out by highly trained and qualified graduate students from the University of Malawi to ensure high quality data. Initially, baseline data was collected to generate the sample size and develop household questionnaires. The study was conducted in multiple stages based on the degree of intricacy nature of the study. Phase one involved the use of a well-structured household CV 'survey questionnaire. Phase two of the study involved a comprehensive review of the secondary data, such as relevant books, internet articles, and government reports. The critical information that was obtained from the reports include social economic and the demography of the communities at Chia lagoon.

### 3.4. Sample and Sampling Design

After the reconnaissance survey, semi-structured and structured household questionnaires and a checklist were framed, pre-tested, and checked for internal validity and reliability. The households for the interviews were purposively sampled based on their proximity to Chia lagoon. The surveys took place in four villages (Ngalauka, Kalimanjila, Mpamantha, and Nkhanga). To calculate the sample size for the study, the following formula was used [15]:

$$n_r = P(1-p)\left(\frac{z}{e}\right)^2 \tag{1}$$

where *n* = sample size and *z* = value from the standard normal distribution reflecting the confidence level (*z* = 1.96 for 95% confidence) of unknown population proportion (*p*). P = 0.5 was used, which assumes maximum heterogeneity (i.e., a 50/50 split) since the proportion of the population was not known [15]. 0.057 margin of error (*e*) was used in order to have a statistically representative sample size with the

highest precision. The calculated sample size was 300. This implies that a household questionnaire was administered to about 300 households around the Chia lagoon.

The households for interviews were selected randomly while using the catalogue that was obtained from the village head in the study area. The questionnaire was designed in English, which is an official language in Malawi. However, during the interviews, the questionnaire was translated into Chichewa (the national language). The questionnaire was administered to the household after seeking accord from the respondents. Five research assistants helped in administering the questionnaire after being trained and pre-testing the questions in a different community. Data was collected for a period of 14 days. The data collection techniques followed high ethical values [23].

### 3.5. Focus Group Discussion (FGD)

A checklist was used to collect information from focus group discussion (FGD). Themes regarding the water resources situation at Chia lagoon were deliberated during the focus group discussion. Eight focus group discussions comprising an average of 12 to 20 members were coordinated by one research assistant. The questions were presented as a guideline while using three approaches, such as resource mapping, institutional analysis, and cause-effect analysis [80]. To ensure the maximum involvements and full ingenuousness of each individual group, the respondents were split in respective to gender and age. FGDs were used to generate the deeper understanding of the past and present situation of water resources in the lagoon.

### 3.6. In-Depth Interviews of Key Informants

A checklist was developed to provide guideline during key informant interviews. Key informants, such as chiefs, local leaders, conservation groups, and elderly residents, were interviewed to explain their experience regarding water resources situation in the Chia lagoon. A 'snowball' technique was employed in the selection of each key informant and locates one or two other possible informants through networks [81]. The snowball sampling techniques works in such a way that key informants are obtained from each informant and the sample size grow with a subsequent interview until the sample size becomes saturated.

### 3.7. Direct Observation

Direct observation was also conducted to generate a rich data set and also depict the situation of water resources in Chia lagoon, observe livelihoods activities, social networks, and governance issues. The idea behind the direct observation was to try to understand the reason why people behave the way they do. Direct observation was done by frequently visiting the communities for an extended period of six months to observe and inquire about their culture, social networks, attitude, governance issues, and social economic activities, and try to link them to the value that they attach to the resources [81].

### 3.8. Willingness-to-Pay Hypothetical Model Approach

The study adopted CV techniques. Individuals were presented with information regarding specific water quality changes in the lagoon. A contingent valuation survey instrument was used to capture the values that could not be captured through economic market instrument [43]. Individual's attitude, perception, and preference regarding water quality changes in the lagoon and its non-market values were elicited [43]. The respondents were typically asked for their WTP in order to determine the extent to which water quality changes have been affecting individuals 'welfare [55,82]. Money was used as a standard measure of a change in risks of being affected by poor water quality in the lagoon and it was defined as either positive or negative payment, which, according to the role of economics [15], holds expected utility constant under different risk levels. *Ceteris paribus,* high risk may imply high WTP amount that is categorically equated to high improvement of water quality and the reduction of risk of being affected by poor water quality [83]. As previously reported by other existing literatures, understanding the risk of being affected by the polluted water in the lagoon may be proven difficult

to comprehend, especially in Malawi, where different groups are at different levels of risks [56]. Additionally, this approach is also proven difficult, especially where water uses across individuals and communities are characterized by high illiteracy [43].

However, in this study, different risk awareness on poor water quality in the lagoon was carried out during the reconnaissance survey, which helped to remove the intellectual problem of understanding and interpreting probabilistic representative of risk especially associated with large group of illiterates [43]. The risk of communities being affected by water pollution in the lagoon were categorized as exogenous, where factors are beyond individuals' control and endogenous, where communities can take action to reverse the situation and reduce undesirable likelihood of unparalleled events, such as floods, disease outbreak, and others from occurring or reducing the cost of event in case it occurs [15,43]. In this study, it was assumed that individual's risk reducing actions is a function of WTP. Economic theory predicts that individuals equate the marginal benefits of self-protection with the marginal cost, as subject to their budget restriction [43]. Hence, WTP provides an indicator of the associated total economic welfare impact, and also the total economic value of the resources [84]. In determining the WTP amount across the beneficiaries, a number of influencing factors were accounted [43]. In other words, WTP was estimated as a function of various influencing factors, such as realizing the gravity of risk, which is determined by exogenous factors (F), self-protection (Pi), income (Yi), risk eversion (Si), and was expressed by the following logistic regression model [43]:

$$WTP_i = \ln\left(\frac{P}{1-P}\right) = \beta_0 + \beta_1 X_1 + \beta_2 \beta_2 + \beta_3 \beta_3 + \dots \beta_n X_n \qquad (2)$$

where $WTP_i$ is a dummy variable (where 0 = positive WTP and 1 = negative WTP), $P$ as a dependent variable of probability 1, the parameter $\beta_0$ is constant and estimates $(\beta_n)$ is the regression coefficient, and $X_n$ is both the endogenous and exogenous risk factors [85]. Endogenous risk factors consist of protective measures that respondents were expected to take and it was controlled through the information collected in the CV survey. The extent of the risk depends on the nature of water quality degradation in the lagoon. The main hypothesis in this study was that WTP for improvement of water quality in the lagoon would be consistent with variations in exogenous risk levels across individual households. However, realizing risk alone could not justify WTP to improve water quality in the lagoon. WTP depends on several factors amongst being socio-economic, demographic, and institutional factors (Table 1).

**Table 1.** Description of explanatory variables used in logistic regression model.

| Explanatory Variables | Definition of Variables | Description of Variables |
|---|---|---|
| GH | gender | Dummy variable where male = 0 and female = 1 |
| AGH | age of household head | Continuous |
| CS | Civil status | Dummy variable where married = 0 and single = 1 |
| LL | Literacy level | Dummy variable where not educated = 0, primary level=1, secondary level = 2, tertiary level = 3 |
| HS | Household size | Continuous |
| LMSWA | Lagoon main source of water for agriculture | Dummy variable where relies on lagoon = 0 and 1 = otherwise |
| LOS | Land ownership | Dummy variable where 0 = has land in the lagoon periphery and 1 = otherwise |

**Table 1.** *Cont.*

| Explanatory Variables | Definition of Variables | Description of Variables |
|---|---|---|
| FLP | Farm in the lagoon periphery | Dummy variable where 0 = farm in the lagoon periphery and 1 = otherwise |
| PIF | Practice irrigation farming using water from the lagoon | Dummy variable where 0 = practice irrigation and 1 = otherwise |
| AFG | Access to food gathering from lagoon waters | Dummy variable where 0 = has access and 1 = otherwise |
| DBLP | Does business in the lagoon periphery | Dummy variable where 0 = do business in the lagoon periphery, 1 = otherwise |
| HALI | Household annual level of income | Continuous |
| HST | Household 'social trust | Dummy variable where 0 = has social trust and 1 = otherwise |
| WRCCESNI | Water Resources Conservation Civic education and social networking involvement | Dummy variable where 0 = has an access Civic education and social networking involvement |
| IT | Institutional trust | Dummy variable where 0 = has institution trust and 1 = otherwise |
| KWDD | Knowledge of water resources degradation and depletion | Dummy variable where 0 = has Knowledge of water resources degradation and depletion and 1 = otherwise |
| AEXTS | Access to extension | Dummy variable where 0 = has access to extension and 1 = otherwise |
| KKWRUR | Knowledge on water resources user rights | Dummy variable where 0 = has Knowledge on water resources user rights and 1 = otherwise |
| AIIWRM | Access to information on IWRM | Dummy variable where 0 = has Knowledge on IWRM and 1 = otherwise |
| TSA | Time the household head has stayed in the area | Continuous |
| DFL | Does fishing in the lagoon | Dummy variable where 0 = does fishing and 1 = otherwise |
| MO | Main occupation | Dummy variable where 0 = Fishing, 1 = Farming, 2 = Traders, 3 = Formal employment, 4 = no occupation |
| DTL | Distance to the lagoon | continuous |

It is very important to note that individuals' risks that are associated with water quality degradation in the lagoon depend on awareness and experiences. In other words, whether poor water quality in the lagoon has negatively affected the health of individual households, livelihoods and income generation depend on awareness and experiences of the particular individual household. It was assumed that individual preferences are likely to be heterogenous towards the improvement of water quality in the lagoon as the result of education level, gender, age [86], size of the household, among other factors. These factors may influence individual's attitude and perception towards the risk of poor water quality, and hence their willingness to trade off the risk reductions against income.

### 3.9. Statistical Analysis

The first question was to understand the perception of the respondents WTP to improve the water quality in the lagoon. This was done by asking the respondents whether they are willing to pay or not without mentioning any specific amount of money. The respondents were requested to be truthful to

their responses when considering their limited income, which can also be used for other important expenses [43]. Following the response from the first question, the respondents were categorized as either positive or negative, protest, or genuine zero. The questions were made simple when considering the high illiteracy in the study area [43]. The exact WTP questions were:

> *Q1. Suppose the authority want to improve water quality in Chia lagoon to reduce the risk of your household being affected by bad water quality and to ensure that everyone in the community benefits from various services rendered by the improved water quality in the lagoon and unfortunately, the authority faces budget deficit and would like to ask you to contribute towards the implementation of this proposed program, would you be willing to pay to improve water quality in the lagoon?*
> *Yes/No (If no, go to question 3)*
> *Q2 If you are willing to pay, can you explain the reasons?*
> *Q3 If you are not willing to pay, what could be the reasons?*

Based on positive WTP response from question 1, the following bid were proposed (100, 150, 200 … 3000 MK). Depending on the first response (Yes/No), the respondents were further asked for their WTP for a follow up bid to which they can again answer (Yes/No). If the respondent fails to answer the first bid, the following bids are either lower or higher. The bid levels were pretested first before being used in the survey. Double bound dichotomous choice (DBDC) was used in this study. The double bound dichotomous choice (DBDC) is the extended version of the single bound dichotomous choice (SBDC) format, where WTP depends on variety of factors Xi, including bid price and unobservable factors that were captured in the error ($\varepsilon i$). In the DBDC format, the respondents are asked a second follow up WP question after the first WTP questions [43]:

$$WTP_i = X_i\beta + \varepsilon_i \tag{3}$$

$$WTP_i^2 = (1 - y)WTP_i + Y\beta + \delta \tag{4}$$

where Y is the parameter reflecting on the starting bid $\beta_i$ and $\delta$ is a shift parameter. Information on WTP intervals is obtained based on DBDC CV approach. The respondents were asked two questions for double intervals [43].

$WTP = \beta^2$ accept both starting bid (B) and follow up bid ($\beta^2$)
$\beta^2 \leq WTP < \beta^1$ reject the starting bid ($\beta^1$) and accept the follow up bid ($\beta^2$)
$WTP < \beta^2$ Reject both bids ($\beta^1$) and follow up ($\beta^2$)

Deriving the probability of observing each of the possible choices, sequence, the jth contribution to the likelihood function was specified as:

$$L_i\left(\frac{WTP_i}{\beta^1}\right) = P_r\left(WTP_i + \varepsilon_{ij} > \beta_i \, WTP_2 + \varepsilon_{2j} \geq \beta^2\right)^{YY} \tag{5}$$

$$P_r\left(WTP_i + \varepsilon_{ij} > \beta_i \, WTP_2 + \varepsilon_{2j} \geq \beta^2\right)^{YN} \tag{6}$$

$$P_r\left(WTP_i + \varepsilon_{ij} > \beta_i \, WTP_2 + \varepsilon_{2j} \geq \beta^2\right)^{NY} \tag{7}$$

$$P_r\left(WTP_i + \varepsilon_{ij} > \beta_i \, WTP_2 + \varepsilon_{2j} \geq \beta^2\right)^{NN} \tag{8}$$

where $WTP_i$ and $WTP_2$ are the means for the first and second bid response and YY = yes and YN = I for yes-no answer, NY = for a no-yes answer, and NN = I for no-no answer. This likelihood function is estimated while using the probit model (cumulative distribution function with zero mean (WTP), correlation coefficient p, the jth contributing to the univariate probit likelihood model) [45]:

$$L_i\left(\frac{WTP_i}{\beta_{\beta^1}}\right) = \varnothing_{\frac{\varepsilon}{\varepsilon_2}}\left(d_{1j}\left(\frac{\beta_1 - WTP_1}{\sigma^1}\right)d_{2j}\left(\frac{\beta_2 - WTP_2}{\sigma^2}\right), d_{1j}d_{2j}\rho\right) \tag{9}$$

where $WTP_{1j} = 1$ if the response to the first question is yes or otherwise, $WTP_{1j} = 1$ if the response to the second question is yes or otherwise, $WTP_{2j} = 1$ if the response to the second question is yes; or otherwise, $d_{1j} = 2WTP_{1j} = 1$ and $d_{1j} = 2WTP_{2j} = 1$

Mean and median WTP were derived, as follows:

$$\text{Mean } WTP = exp\left(\frac{\overline{X}\hat{\beta}'}{\hat{\beta}_0} + 0.5\hat{\sigma}^2\right)$$

$$\text{Median } WTP = exp\left(\frac{\overline{X}\hat{\beta}'}{\hat{\beta}_0}\right)$$

where $\overline{x}$ is a k + 1 row vector of mean value of the explanatory variable, including 1 for constant term, $\hat{\beta}'$ is ak-1 column vector of estimated coefficient and $\hat{\sigma}$ is the estimated variance [87]. STATA version 11 was used to calculate the specified confidence intervals around the mean and median, as recommended by Jeanty [88].

## 4. Results

### 4.1. Characteristics of the Respondents

The study targeted a total sample size of 300 households. However, 240 questionnaires were administered, which represented 80% of valid response rate. Table 2 shows the summary of the results with descriptive statistics, such as variable name, average value, minimum values, maximum values, and standard error.

**Table 2.** Socio-economic and demographic characteristics of the respondents (N = 240).

| Factors | Information Category | Value | Percent | Mean ± STD Error | Min-Max |
|---------|---------------------|-------|---------|------------------|---------|
| Household Head Main Occupation | Fishing | 154 | 64 | 1.69 ± 0.01 | 1–4 |
| | Farming | 48 | 20 | | |
| | Traders | 22 | 9 | | |
| | Formal employment | 17 | 7 | | |
| Level of education | No education | 133 | 66.5 | 0.3 ± 0.03 | 0–2 |
| | Primary education | 54 | 27 | | |
| | Secondary | 13 | 6.5 | | |
| | University | 0 | 0 | | |
| Household size | Less than 4 | 56 | 28 | 7.1 ± 0.65 | 3–12 |
| | More than 4 | 144 | 72 | | |
| Civil status | Married | 156 | 78 | | |
| | Single | 44 | 22 | | |
| Level of income | Less than 1US$ | 178 | 74 | 1.3 ± 0.02 | 0.1–4 |
| | More than 1US$ | 62 | 26 | | |
| Gender of household head | Male | 202 | 82 | 0.33 ± 0.03 | 0–1 |
| | female | 38 | 18 | | |
| Age of household | | 240 | | 38 ± 0.19 | 21–88 |
| Distance to FSTD (km) | | 240 | | 9.32 ± 0.61 | 0.5–20 |

As seen from Table 2, the average age of households was 38. However, the study conducted in Ghana reported age distribution within the range of 41–50 years old. The systematic literature review showed that approximately 73% of the population in Malawi is below the age of 35 years [89,90]. Kayamba-Phiri [91] also noted that the population of fishers in the Western Shore of Lake Malawi is youthful. During field observation, it was noted that the gender disparity is very high among Chia lagoon communities. For instance, the majority (82%) of respondents recorded in the study were male. Birdir et al. [63] reported gender balance of 58% males and 42% females. On the other hand, Kopa [89] reported 73% male respondents in Nkhotakota. Although matrilineal systems that were practiced by communities around Chia lagoon give high privileges for land to women, unequal decision

making in the management of natural resources in the area exist. Chiwawula et al. [92] had similar observations among the Lake Chilwa communities. Chiwawula et al. [92] noted that decision making at the household level on food and income was dominated by men. Nagoli and Chiwona-Karltun [75] also noted that fishing activities were dominated by men in Lake Chilwa. A systematic literature search shows that the management of natural resources, such as water, has been multifaceted in Malawi due to the fact that Malawi policies give priority to male in the management of natural resources overlooking the role of women [39].

It was further interesting to note that 78% of the respondents were married, with other 22% being single, widowed, divorced, or separated. About 72% had a family size of above 4. The average family household size was reported to be 7. About 66.5% had not attended school meaning that about 66.5% of the respondents could not read nor write. Conversely, Malawi demographic and health survey and World Bank reports indicate a slightly higher literacy level (69.7%) in the Nkhotakota district [12] and 62% in Malawi [93] than what the study reported. The study further recorded the average level of income of 1.3US$. The maximum and minimum income level ranges from 0.1 to 4US$, and about 74% of the respondents had income of less than U$1 per day. The study findings agree with Limuwa et al. [23], who reported high income disparities among the fishing commutes in Nkhotakota district. The main occupation and source of livelihood for Lake Shore communities in Malawi is fishing [23,94]. The study found that about 64% of the respondents depend on fishing as the main occupation. The other occupations that were reported in the study were farming (20%), traders (9%), and formal employment (7%). Makwinja et al. [15] also noted that fishing and farming were the major socio-economic activities for Lake Chilwa communities. Similarly, Limuwa et al. (23) showed that fishing (90%), farming (6%), and operating small business (4%) were the main livelihood sources among communities in the Western Shore of Lake Malawi. Makwinja [16] had similar observation in Nkhotakota, Malawi. Government of Malawi report also indicates that communities in the Western Shore of Lake Malawi are predominately fishers [95]. These results indicate the significance of water resources in Chia lagoon to the socio-economic and livelihoods of the surrounding communities. Table 3 shows the descriptive statistics of institutional variables.

**Table 3.** Descriptive statistics of institution factors.

| Factors | Information Category | Value | Percent | Mean± STD Error | Min-Max |
|---|---|---|---|---|---|
| Household 'social trust, | Yes | 95 | 39.6 | 0.84 ± 0.01 | 0–1 |
| | No | 144.96 | 60.4 | | |
| Water Resources Conservation Civic education and social networking involvement | yes | 113 | 47.5 | | |
| | No | 127 | 52.5 | 0.43 ± 0.04 | 0–1 |
| Do you trust that government is capable of implementing the water resources conservation program | yes | 162 | 67.3 | | |
| | No | 78 | 32.7 | 0.76 ± 0.00 | 0–1 |
| Does the water resources problem directly affect the household | Yes | 176 | 73.3 | | |
| | No | 64 | 26.7 | 0.31 ± 0.06 | 0–1 |
| Influence of extension contact | yes | 143 | 59.4 | | |
| | No | 97 | 40.6 | 0.52 ± 0.03 | 0–1 |
| Does the household have knowledge on water resources user rights | yes | 112 | 46.5 | | |
| | No | 128 | 53.5 | 0.14 ± 0.00 | 0–1 |
| Household has access to information about IWRM | yes | 105 | 43.6 | | |
| | No | 135 | 56.4 | 0.12 ± 0.02 | 0–1 |
| Does any water resources conservation committee around the lagoon get funding from local or external organization | Yes | 91 | 38 | | |
| | No | 149 | 53.5 | 0.08 ± 0.00 | 0–1 |
| Does household have knowledge of water resources degradation or depletion at the lagoon | Yes | 157 | 65.3 | | |
| | No | 83 | 34.7 | 0.12 ± 0.00 | 0–1 |

Results from quantitative analysis (Table 3) revealed that the majority of the households (61%) lost trust in the existing natural resources conservation committees. Similarly, Nagoli and Chiwona-Karltun [53], in their analysis of reliability and trustworthiness of the institutions in Lake



Chilwa, found that the elected committees were not as effective in knowledge sharing and resource governance as compared to the informal networks. On the contrary, the study revealed that the majority (67%) of the households believed that government is capable of implementing the water resources conservation program at the lagoon. The study further revealed that the majority (53.2%) of the respondents had poor access to water resources conservation civic education and social networking, despite the majority (65.3%) being aware of the problem of water resources degradation and the depletion in the lagoon. However, Nagoli and Chiwona-Karltun [75] emphasized the significance of socio-networks, as it both formally and informally connects people in the society. According to Lee [96], a social network is defined as a form of social coordination among the individuals and organizations. Similarly, Folke [97] identified social networks as an important denominator in uniting different stakeholders to successfully deal with natural resources problems in the society. It is very apparent that poor social network and civic education could be a contributing factor to water quality degradation in Chia lagoon.

*4.2. Analysis of the Households' WTP*

Table 4 shows the distribution of households WTP responses. In this study, the household WTP responses were categorized as protest, positive, and genuine zero. The protests were distinguished from genuine zero by asking respondents why they were not willing to pay for the program. Those who responded that they had low level of income and could not pay for the program were categorized as genuine zero. Other responses were categorized as protest.

**Table 4.** Distribution of willingness-to-pay (WTP) responses.

| | Income Reported | | |
|---|---|---|---|
| **WTP** | **No** | **Yes** | **Total** |
| Protest | 32 (0.50) | 22 (0.12) | 54 (0.225) |
| Positive | 15 (0.23) | 138 (0.78) | 153 (0.64) |
| Zero | 16 (0.25) | 17 (0.09) | 33 (0.14) |
| Total | 63 (1.00) | 177 (1.00) | 240 (1.00) |

Table 4 shows that 22.5% of the respondents protested the idea of paying to improve water quality at the lagoon. On the contrary, 64% of the respondents were positive, while genuine zero made up of 33%. Table 5 shows the results of analysis of the WTP amounts.

**Table 5.** Analysis of WTP amount (MK).

| Parameter | Number | Mean | Median | Mode | Minimum | Maximum | Sum |
|---|---|---|---|---|---|---|---|
| Amount/month | 138 | 7870.455 | 2040 | 2040 | 696.825 | 81,697.23 | 456,684.435 |

MK733.5 = 1 US$ as of December 2016.

A total of 138 out of 240 respondents (57.4%) were willing to pay to improve water quality in Chia lagoon (Table 5). However, Birdir et al. [65] reported 92% WTP amount to see the improved beaches in Mersin, Turkey. Liu et al. [98] reported the 53% rate of WTP for improved air quality in China. In New Zealand, Omwenga [99] reported 25.8% in CV case studies. Barnnet et al. [99] on the hand achieved the response rate of 47.3% in Australia. The lowest willingness to pay value recorded from the study was MK696.825 per month and maximum was MK81697.23. The median was recorded with the assumption that there may be extreme values in the data set, which, in this case, median could be more useful than mean. On the other hand, mode was recorded to display the most frequent number in the data set. The median recorded from the study was MK2040, while the mean was MK 7870.455 and the mode MK2040. Figure 1 shows a linear trend line fitted the probability of respondents' series. Linear regression model **y = –0.5857x + 27.651** showed that the rate of respondents increased with the decrease in the rate of WTP amount.

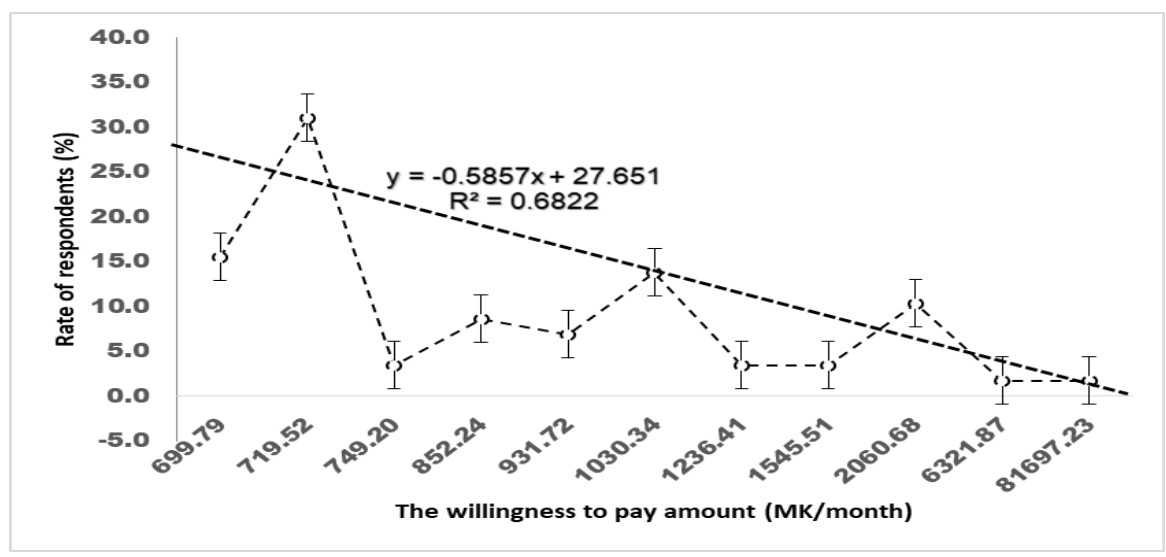

**Figure 1.** Analysis of WTP amount (MK).

The coefficient of linear regression model ($R^2$) was 0.6822, which suggested that 68.22% variation in the rate of respondents were influenced by the rate of WTP amount. The response rate randomly varies from one WTP amount to another. The rate of response dropped from 31% to 15.5% with a decrease in the WTP amount from MK 748.14 (US$1.02) to MK 696.83 (US$0.95). On the other hand, the response frequency was lowest at MK 81,697.23 (US$111.38) WTP amount. Overall, the study revealed that the respondents were willing to contribute, the monthly range of MK 696.83 (US$0.95) to MK 81,697.3 (US$111.38) per individual and monthly average of MK7870.45 (US$10.73) per individual generating aggregate values that ranged from MK 6,689,568 (US$9,126.29) to MK 784,294,080 (US$1,069,978.28) annually and on average MK75,556,320 (US$103,078.20) per annum (*ceteris paribus*).

Table 6 present the reasons for positive and negative WTP responses. As seen from Table 6, 27% of the respondents were less satisfied by the services that were rendered by water resources in the lagoon. On the contrary, 40% of the respondents were satisfied and 17% very satisfied. The empirical evidence from the study indicates that the rate of WTP amount increases when the communities are happy with the water quality in the lagoon. This explains that local authorities need to establish water resources management coordination committee of which its primary purpose is to formulate relevant development policy and strategic plan for lakes, wetlands, lagoons, and river basins. The authority must further empower the communities around the lagoon to take charge of the management so as to improve water quality and quantity in the lagoon. This effort can stimulate a household's WTP and take positive contribution to the protection of water resources in the lagoon. Again, deliberate effort must be made to maximize the household income if the household is to respond to high WTP amount, thus bringing the beneficiaries back to the original level of utility. As seen from Table 6, about 33% were willing to pay, because water resources in the lagoon improved their income.

During FGD, some of the households said "Almost everyone in this community access drinking water from the lagoon and very few have access to drinking water from public utilities". Evidently, women were spotted during the field observation fetching water from the lagoon for domestic purposes. It was further noted that about 50.5% of the respondents were willing to pay, because they believed that the proposed program would improve water quality, increase fish biodiversity, mitigate climate change impact, and consequently improve their livelihood. However, 16% of the respondents were not satisfied by the services that were rendered by the water resources in the lagoon. During focus group discussion, some of the respondents expressed that "if Chia lagoon dries up, we will have large fertile land for farming". About 42.4% of the respondents who were not willing to pay expressed that it was government responsibility to improve water quality in the lagoon (42%). On the other hand, 51% disagreed with the program and 21% expressed that the program could not improve their income, and

hence no need to pay. Other respondents reserved their comments, but expressed that they were not willing to pay for the proposed program. These categories of respondents were classified as protests. On the other hand, 4% of the respondents explained that they could not pay for the program due to low level of income.

**Table 6.** Reasons for positive and negative WTP responses.

| Factors | Information Category | Value | Percent | Mean ± STD Error | Min-Max |
|---|---|---|---|---|---|
| Ranking Household satisfactions by the services rendered by Chia lagoon water resources | Not satisfied | 38 | 16 | 2.53 ± 0.02 | 1–4 |
| | less satisfied | 65 | 27 | | |
| | satisfied | 96 | 40 | | |
| | Very satisfied | 41 | 17 | | |
| WTP | Positive WTP | 138 | 57 | 0.5 ± 0.02 | 0–1 |
| | Nagative WTP | 102 | 42.6 | | |
| Reasons for not WTP | Has no money | 9 | 4.5 | 3.47 ± 0.087 | 1–5 |
| | Government should pay | 42 | 21 | | |
| | Disagree with the program | 51 | 25.5 | | |
| | Water resources at Chia lagoon has no impact on Household income | 42 | 21 | | |
| | Other reasons | 56 | 28 | | |
| Reasons for WTP | Improves household income | 33 | 16.5 | 2.4 ± 0.08 | 1–5 |
| | Increase lagoon biodiversity | 101 | 50.5 | | |
| | Improves water quality | 29 | 14.5 | | |
| | mitigate impact of climate change | 8 | 4 | | |
| | Other reasons | 29 | 14.5 | | |

### 4.3. Analysis of Factors Influencing WTP

Demographic, socio-economic, and institutional factors were tested for multicollinearity before introducing them into logistic regression model (Supplementary File S3). The results showed that multicollinearity was not a problem. None of the explanatory variables were strongly correlated or overlapped, which suggested that the explanatory variables were fit to be presented in the logistic regression model. Table 7 shows the logistic regression estimates of the probability of the respondents' WTP.

The R-squared values are expected to be constrained to the range of 0–1 (0–100%), with higher values indicating the better model fit, according to principles of econometrics [100,101]. In this case, the Cox and Snell (1989) were used to test the model's power of prediction. However, a modified Cox and Snell was also used to allow R-squared to take on values in the full range of zero to one, making pseudo R squared more compatible to a conventional R-squared statistic. Table 7 showed Cox and Snell (1989) R-squared (0.54), and Nagelkerke (1991) R-squared (0.73) statistics. Hosmer and Lemeshow (1989) Chi-square value was 5.44 and non-significance as ($p > 0.05$). The overall analysis indicate that the logistic regression model explained a substantial amount of variance in the choice of the respondents' WTP. Hosmer and Lemeshow, (1989) goodness of fit test was greater than 0.05 and was non-significant at 0.05, which indicated that the models were a reasonably good fit to the data and, therefore, good overall model fit. Logistic regression was conducted to determine whether demographic, socio-economic, and institutional factors are associated with local communities' attitude of WTP. As shown in Table 7, these demographic (gender, age, literacy level), social-economic (land ownership, main agriculture water source, and income), and institutional (civic education and social network, extension, institutional trust, household socio trust) factors had a significant ($p < 0.05$) effect on WTP.

**Table 7.** Effects for the best fitted logistic regression model of demographic, socio-economic, and institutional variables.

| Explanatory Variables | β | S. E | Wald | Sig |
|:---:|:---:|:---:|:---:|:---:|
| GH | −1.19 | 0.71 | 2.81 | 0.02 * |
| AGH | −2.77 | 0.85 | 10.63 | 0.00 ** |
| CS | −0.15 | 0.75 | 0.04 | 0.85 ns |
| LL | 2.31 | 0.83 | 7.52 | 0.01 * |
| HS | −1.15 | 0.71 | 2.66 | 0.10 ns |
| LMSWA | 5.35 | 0.7 | 7.43 | 0.01 * |
| LOS | −1.93 | 0.82 | 5.61 | 0.02 * |
| FLP | 1.12 | 0.12 | 0.04 | 0.85 ns |
| PIF | 0.45 | 0.28 | 0.12 | 0.72 ns |
| AFG | 1.65 | 0.29 | 1.63 | 0.06 ns |
| DBLP | 0.42 | 0.64 | 2.67 | 0.08 ns |
| HALI | 1.63 | 0.28 | 5.12 | 0.02 * |
| HST | −3.21 | 0.8 | 2.31 | 0.03 ns |
| WRCCESNI | 1.83 | 0.85 | 6.97 | 0.02 * |
| IT | 5.74 | 0.76 | 8.95 | 0.01 ** |
| KWDD | 0.28 | 0.88 | 0.1 | 0.06 ns |
| AEXTS | 2.47 | 0.83 | 8.83 | 0.00 ** |
| KKWRUR | 0.8 | 0.82 | 1.53 | 0.06 ns |
| AIIWRM | 1.52 | 0.75 | 4.1 | 0.043 ns |
| TSA | 0.28 | 0.88 | 0.1 | 0.06 ns |
| DFL | 0.26 | 0.43 | 2.31 | 0.13 ns |
| DTL | 0.21 | 0.63 | 2.31 | 0.13 ns |
| Constant | −2.62 | 0.82 | 10.25 | 0.00 ** |

Hosmer and Lemeshow test, Chi- square = 5.44 (df = 8), P = 0.71 -2log likelihood = 59.1%, Note: Nagelkerke R Square = 0.73, Cox & Snell R Square = 0.54. ns indicates not significant while ** and * indicate significance at 0.01 and 0.05 probability level of Confidence.

## 5. Discussion

During the exploratory survey, it was noted that water quality degradation in Chia lagoon is a major public concern. For instance, some of the households expressed the following experiences: "...In early around 1970s, Chia lagoon had clear water throughout the year, however, in recent times, the quality of water in the lagoon has severely deteriorated...." A synthesis of exploratory secondary data extracted from relevant documents, such as books, publications, journal articles, and reports conform to the household's experiences. During focus group discussion, the communities explained that changes in water quality in the lagoon were attributed to anthropogenic activities [102] and climate change [103]. It was noted that the situation experienced at Chia lagoon is in line with what has been reported elsewhere by different scholars. In Kenya, Muriuki et al. [104] noted that rapid population growth, deforestation, diminishing land holdings, erratic rainfall patterns, and conflict in water use are among the significant factors influencing watershed degradation, which leads to the loss of wetlands in some areas, increasing water pollution, and a decrease in water levels, which in turn negatively affected the aquatic ecosystem. In Ethiopia, Wolka et al. [105] observed that increased flooding and newly formed rills and gullies in watershed resulted in the loss of Cheleka wetland. In Rwanda, the anthropogenic activities in the watershed resulted in the degradation of Rugez Marsh [106].

The results from field observation established that prolonged turbidity, siltation of the lagoon, and an increase in invasive alien species were the major indicators of water quality degradation. It was noted that the opening new land for agriculture, cultivation on steep slopes and stream banks, poor farming practices, felling of trees for wood, and setting bush fires that destroy or degrade valuable vegetative cover significantly contributed to the serious problems of water runoff and the loss of top soil, especially in the upper reaches of the watershed, leading into sedimentation of Chia lagoon [19]. It was further noted that there is major concern among government, NGOs, and local communities on the current state of water quality. Moldan [107] observed that insufficient availability

of water to meet all demands, including the environmental flow requirements, has significant impact on the household's WTP. The local communities have strong intention to improve the water quality. Communities' intention was linked to the notion of neoclassical economic principles [108]. The interests of local communities to improve water quality was equated to their WTP.

Table 4 shows the various responses made by the communities towards the WTP amount to improve water quality at Chia lagoon. A comparison of the distributions between the groups with respect to the type of proposed WTP amount made provides a clear picture, which suggests that there is a relationship between income non-response and the WTP amount proposed. Reading the column percentage that is indicated in Table 4 shows that respondents who refused to answer the income questions appeared to have a higher tendency to report genuine zero and protest WTP program as compared to those who report their income. Figure 1 showed a linear trend line fitted response series that indicated a slight downward slope. The explanation for this observation was due to the influence of the respondent's decision to participate in the water quality improvement program in Chia lagoon. The response rate seems to randomly vary from one WTP amount to another. Contrary to the theoretical expectation, the response rate dropped from 31% to 15.5% with a decrease in the WTP amount from MK 748.14 (US$1.02) to MK 696.83 (US$0.95). In reality, it was expected that the respondents offered MK 696.83 (US$0.95) WTP amount to find it easy to positively respond, as the financial burden associated with 'yes' would be low. On the other hand, the response frequency was the lowest at MK 81,697.23 (US$111.38) WTP amount. The probable explanation for the observed low response frequency in the bid amount of MK 696.83 (US$0.95) could be; the respondents given that this amount could not believe that water quality improvement program in the lagoon could be financed through such a small contribution, and hence could not take the survey serious. Carson et al. [109] claimed that the respondents who face the low bid that seem unrealistic are likely to replace this bid with 'expected cost' and respond accordingly, resulting in higher proportion of 'no' to the bid than one would otherwise expect.

Figure 1 further indicated that the proportion of respondents answering 'yes' (the probability of answering 'yes') to the question of willingness to pay decline as the bid amount increases. Wheeler and Damania [110] reported a similar pattern in 'yes; responses as the bid amount increases. Economic theory predicts a downward trend in consumer response to an increase in price of normal goods or services [110]. Therefore, it may be concluded that the respondents in this study behaved in a manner that is consistent with economic theory. It was also interesting to note that the proportion of responding 'yes' was higher than that of responding 'no' to the low bid amount of MK 696.83 (US$0.95). This means that respondents were more likely to say 'yes' than 'no' at a lower WTP amount. Although the FAO [111] report revealed that it is very difficult to set specific price on how much each water user should pay to conserve the resources, the study showed that at MK 718.83 (US$0.98) WTP amount, water quality in Chia lagoon can be improved.

The study also estimated the aggregate values of WTP amount. The findings showed that the respondents were willing to contribute, the monthly range of MK 696.83 (US$0.95) to MK 81,697.3 (US$111.38) per individual and the monthly average of MK 7870.45 (US$10.73) per individual. The interviews were conducted in four villages found along Chia lagoon periphery with approximately 800 households. Assuming that all of the households were willing to pay to improve water quality in the lagoon, aggregate annual values ranging from MK 6,689,568 (US$9126.29) to MK 784,294,080 (US$1,069,978.28), and on average MK 75,556,320 (US$103,078.20) (*ceteris paribus*) can be generated. The study findings agree with what has been reported in literature. For instance, Venkatachalam and Jayanthi [112] in Chennai City, India found that households were willing to pay a maximum lump-sum amount of Rs. 2096.59 per annum for improvements in the quality of Pallikaranai marshland. Oglethorpe and Miliadou [113] also reported that the sustainable management of Lake Kerkini in Northern Greece could be justified on the basis that respondents were willing to pay, on average, £15.24 per person per year, generating an aggregate value of £23.3 million per year. Again, Zhongmin et al. [114] estimated the total economic value of restoring ecosystem services in the Ejina region in China and

obtained a present value of aggregate willingness to pay of US\$ 6.67 million per year. Loomis et al. [115] further estimated the total economic value of restoring ecosystem services in an impaired river basin and reported a household mean WTP estimate of US\$252 per annum and aggregate annual value of US\$ 70 million. In Malawi, Zuze, [116] estimated the monthly mean household WTP to conserve Lake Chiuta's biodiversity to be at MK 325.86 (\$0.91), whereas the monthly aggregate WTP was MK3.3 million (\$9178.69), with an annual aggregate WTP of \$11.0M.

The logistic regression analysis shown in Table 7 demonstrated a significant ($p < 0.01$ or $p < 0.05$) relationship between demographic (Gender, age, literacy level), social-economic (Land ownership, main agriculture water source, and income), and institution (civic education and social network, extension, institutional trust, and household social trust) factors and WTP. It was found that demographic variables, such as age of the household head, was negative, with a correlation regression coefficient of –2.77 and Wald of 10.63 and statistically significant at $\alpha = 0.05$, while the relationship between the dependent variable and the level of literacy was positive with a correlation regression coefficient of 2.31, Wald of 7.52, and statistically significant at $\alpha = 0.05$. Halkos and Matsiori [117] noted that the age of the household had a negative regression coefficient, which explained that order people were not able to contribute much due to several factors, such as high expenditures on food and household health, high economic dependence, and others. The results of this study are in line with many previous CV studies [118,119]. However, critical analysis of the previous literature shows that age had both a negative and positive effect on peoples' WTP. For instance, Mezgebo and Ewnetu [120] found that the respondents aged above 50 years were less willing to pay for improved water services in Mutale Local Municipality, Zimbabwe. On the contrary, Harun et al. [121], in a study done in Iraq, found that older people were willing to pay more for water irrigation than the young farmers.

Education is widely considered to be the most important form of human capital [122] and it can significantly influence the communities WTP. It was interesting to note that education had a positive regression coefficient. Halkos and Matsior [116] earlier stablished that education has an impact on WTP. Kanyoka et al. [122] also found that the level of education had an influence on WTP in South Africa. Previous published studies and theories [123] also supported the present findings. Theoretically, people with higher level of education are expected to understand better than those who are not well educated [117].

The socio-economic variable between main source of water for agriculture **(MSWA)** and WTP had a positive relationship. The regression model was significant at $\alpha = 0.05$ and had a coefficient of 2.35 and Wald of 11.43. A positive relationship between the probability of the respondents was noted in the level of income (LI) and WTP. Regression coefficient was 3.06, Wald of 17.80, and it was statistically significant at $\alpha = 0.01$. On the contrary, the relationship between the probability of the respondents 'WTP and land ownership was negative. The regression coefficient was –1.93, Wald of 5.61, and it was statistically significant at $\alpha = 0.05$. Johnson et al, [124] reported that the decisions on how to manage water resources in the lagoon could be based on the location of land used for agricultural production or settlements. Again, Leeworthy and Bowker, [125] summarized the linkages between the economy and the environment. The model explained that the actual conditions relating to the quality and quantity of water resources in wetlands are important factors that determine the individual's perception towards the conservation. This explains that the level of demand of water resources economic value in terms of agriculture significantly influence the individual's WTP. Arouna and Dabbert, [126] showed a positive correlation between the level of income and WTP for water supply improvement in Benin. Similar results were also reported by Mezgebo and Ewnetu [120] in Nebelet town, Ethiopia. Previous studies also indicate that the increase level of household income could consequently shift the demand curve for improved water quality to the right, suggesting that the households would have better chances of maximizing utility [122,127,128]. Halkos and Matsiori [117] also reported that, as the level of income increases, people would be more willing to pay. Others studies using meta-analysis data had similar findings [129].

Institutional factors, such as water resources conservation civic education and social networking involvement (WRCCESNI), access to extension services (AEXT), institutional trust (IT), and house socio trust (HST) were statistically significant, at $\alpha = 0.01$. It was found that households who had access to water resources conservation civic education through conservation agriculture programs were more WTP in order to improve the water quality in the lagoon. Similarly, the households who were more connected to social networks, such as electronic media, were easily convinced to pay towards the program. The study findings concur with Agudelo [130] who observed that, as the demand for water resources increases to their availability, the communities are more willing to pay towards the conservation program to sustain the resources. Faraji and Mirdamadi [131] had a similar observation among the apple producers in the Damavand area. Other authors also previously mentioned that extension services play a critical role in facilitating linkages between the communities and other relevant sectors, such as government departments, private sectors, non-governmental organization, research institutes, and education centers [132]. A similar observation was made by Mbo'o-Tchouawou and Colverson [133] in rural population in Kenya. Kapanda et al. [134] reported that the lack of extension staff was an important adoption problem. Paris, [135] further observed that the success and failure of improved integrated crop-animal technology depended on the availability of information related to social economic impacts of such interventions on rural communities. Nwankwoala [136] argued that the challenges that are posed by degradation of wetlands could be better understood by the communities if they are properly equipped through awareness for them to understand the economic value that is attached to the resource.

Chia lagoon hosts a diverse group or organizations operating at multiple scales within the entire lagoon socio-ecological system. During the exploratory survey, it was noted that these organizations are entrusted with the responsibility of facilitating decision making process under the co-management approach. Evidently, organizations, such as faith-based organizations (FBO) and Community Based Organization (CBO), directly work with the communities. Other organizations, such as government departments, provide services to the communities, but they are not physically present in the area. Backed by their sectoral policies, some of these organizations formed various natural resources governance structures, such as community policing committees, Beach Village committees (BVC), Fish conservation committees, and Natural Resources Conservation Committees to stimulate effective management of natural resources in the lagoon. It was noted that the effective coordination of water governance approach at multiple scale requires institution trust. Institution trust refers to the level of trust that the communities have in a certain institution governing the natural resources conservation programs [137]. In this study, institution trust was examined in relation to the Chia lagoon water quality situation. Theoretically, institution trust is expected to significantly influence WTP [138]. The more trust that communities have in water quality management institutions, the higher the WTP amount. The regression coefficient of institutional variable was positive, suggesting that high institution trust means the high acceptance of WTP approaches. In other words, the WTP proposal enacted by local authorities, policy makers, and fund management managers of local organization could be successful if the communities have high trust in such institutions. Likewise, Cvetkovich and Winter [139] noted that the high level of institutional trust produces a positive influence on the acceptance of environmental policy.

Household socio trust had a negative regression coefficient. Household socio trust refers to the level of trust that the communities have in the individuals' heading the local institutions governing the natural resources conservation programs. During the study it was noted that members of local governance structures, such as Beach Village Committees, Village Development Committees, Community Policing Committees, and Area Development Committees were weak due to low incentives of their members. It was further pointed out during FGDs that positions, such as chairperson, secretary, and treasurer who were regarded as a symbol of high status and could spur interest among the communities to manage water resources in the lagoon were corrupt. Other members pointed out that these committee members were donor-driven and their interest to improve water quality in the lagoon were driven by

the monetary benefits from donors rather than the will of the people. Nagoli and Chiwona-Karltun [75] had similar observation among the lake Chilwa communities.

The empirical findings of this study show that the impetus behind WTP is mostly based on individual's insight and anticipation of future improved water quality in the lagoon. A number of studies have similar explorations on how water quality in the natural ecosystems influence peoples' WTP [115,123]. This study offers additional outlook to those actions while considering user's preferences and demands. It contributes to the development of more holistic approach to the economic valuation effort of services rendered by water resources in the natural ecosystems. Particular emphasis was given in exploring the attitude of respondents to find out which demographic, socio-economic, and institutional decisive factors affect communities' insights towards WTP to improve water quality in the lagoon. This study provides very useful data to the decision and policy makers to determine the best optimal management strategy of water resources in the lagoon. Evidence that was generated from this study indicate that a great number of respondents were willing to pay for improvement of water quality in the lagoon.

## 6. Conclusions and Recommendation

The findings from this study revealed that the price that was employed on water quality improvement could be substantial and could be efficiently used in water quality management in the lagoon. With this study, it was noted that the estimate can be applied as a fixed price for conservation and improvement of water quality in the lagoon. Again, demographic, socio-economic, and institutional characteristics distinctively affected the perceptions of respondents towards WTP, which suggested that water resources management in the lagoon must aim at balancing the water quality protection and economic efficiency. In other words, decisions regarding future management of water quality in the lagoon must focus on various services under which the resources render to the communities. The study findings have policy implications for local government. Strategies that aimed at managing water resources in the coastal lagoons, lakes, rivers, wetlands, and other natural ecosystems need to consider this kind of study for effective implementation. Better understanding of delicacy of water, WTP amount, and their influencing factors could eventually help to better frame water resources conservation policies and their enactment.

## 7. Limitations

This study was designed based on Chia lagoon natural ecosystem. Therefore, it is imperative that future research should take consideration of other natural ecosystems. For instance, research focusing on investigating how much respondents would like to pay and estimation of total economic value of the natural ecosystem, such as Lake Chilwa, Lake Malombe, Shire River, and other important natural ecosystems in Malawi is very crucial for policy makers.

**Supplementary Materials:** The following materials are available online at http://www.mdpi.com/2071-1050/11/17/4690/s1, File S1: Household survey questionnaire, File S2: CV survey questionnaire, File S3: Result of descriptive statistics and Correlation coefficient of factors affecting the WTP.

**Author Contributions:** R.M. currently studying PhD in Water Resources Management at Addis Ababa University, Ethiopia designed the research, collected data, analysed and developed the manuscript. I.B.M.K. (PhD) and C.C.K. (PhD) from University of Malawi, The Polytechnic were involved significantly at each stage of the manuscript writing, field scoping and reviewing the study tools.

**Funding:** This research received no external funding.

**Acknowledgments:** The authors wish to acknowledge University of Malawi, The Malawi Polytechnic for making this study possible. Many thanks should also go to the communities for the lively participation during the data collection.

**Conflicts of Interest:** The authors declare that there is no conflict of interest.

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
