# Peer review of "Determinants and Values of Willingness to Pay for Water Quality Improvement: Insights from Chia Lagoon, Malawi"

_sustainability, doi:10.3390/su11174690_

Round 1

Reviewer 1 Report

Dear Authors,

I read your work with great interest. My overall impression is that you can improve greatly your paper if you frame it within a more comprehensive analysis, and explanation of the social and institutional background. I had the impression that you were blindly applying an analytical fwk, surely very sound and interesting, but without discussing enough the contextual factors that were more relevant to the specific situation of the Chia lagoon. I felt in many sentences that your approach missed out many other dimensions that should be relevant to evaluate the willingness to pay. Starting to analyse what is needed, who should pay, and what is the level of responsibility, and also the affordability of the local communities, seems paramount in my view.

I think you may improve your manuscript, and include some relevant information that will help the readers to understand better what is the context of Chia lagoon, what are the institutional gaps, and who are the main responsible parts for the bad water quality. A more holistic analysis can help your statistical method to be useful, if you also include some recommendations to the policy level.

I also recommend that you avoid using generic references to economic theory without appropriate justification, since the policy measures that can eventually incentive some local contributions cannot be designed without a deep understanding of the social and environmental conditions.

I annex a word doc with some notes. My apologies for not using the PDF version, which I could not edit, probably due to my own computer difficulties, but I hope the copy I did (that lost the formatting, but contains all the lines numbers) can be enough for you to follow my notes.

I wish you all the best with your next steps, and I think this will be a very useful and relevant work in the future, and a potential support for a better water policy in Malawi.

The Reviewer

Author Response

1.      I read your work with great interest. My overall impression is that you can improve greatly your paper if you frame it within a more comprehensive analysis, and explanation of the social and institutional background. I had the impression that you were blindly applying an analytical framework surely very sound and interesting, but without discussing enough the contextual factors that were more relevant to the specific situation of the Chia lagoon. I felt in many sentences that your approach missed out many other dimensions that should be relevant to evaluate the willingness to pay. Starting to analyse what is needed, who should pay, and what is the level of responsibility, and also the affordability of the local communities, seems paramount in my view.

Response: Thanks for your constructive comments. WTP has been written in full. The statement ‘Water resources in the natural ecosystems is a vital global economic resource’ has been revised to ‘Water resources in the natural ecosystems are considered as a vital global social-economic and environmental goods’. The statements ‘Economic theory explains that individuals have preferences [18]. Their preferences are linked to the willingness to pay (WTP)’ have been removed and the context has been restructured. Over all, the paper has been revised and relevant text have been added in the section of introduction and conceptual framework. A conceptual framework has been revised in response to the paper objectives. The results have further displayed who is supposed to pay and estimation have been made on WTP amount.

2.          I think you may improve your manuscript, and include some relevant information that will help the readers to understand better what is the context of Chia lagoon, what are the institutional gaps, and who are the main responsible parts for the bad water quality. A more holistic analysis can help your statistical method to be useful, if you also include some recommendations to the policy level?

Response: The suggestion has been taken and some relevant information has been added to help the leader to have general overview of water quality situation at Chia lagoon. We have also demonstrated who is responsible for bad water quality and what the community response is. The last section of the recommendation has demonstrated a policy direction.

3. I also recommend that you avoid using generic references to economic theory without appropriate justification, since the policy measures that can eventually incentive some local contributions cannot be designed without a deep understanding of the social and environmental conditions

Response:The suggestion has been taken and the context has been revised and appropriate justification have been backed up with relevant literature.

Reviewer 2 Report

I think it is an interesting paper, but lacks of a well developed methodology. For example, the authors describe unnecessarily the well known logistic regression model (a footnote or a reference would be enough), whilst variables (and their meaning) used in regressions are not explained. The survey sample is not enough big (but this could be logical f the geographical area is difficult to access -I don't know about this issue). Questionnaire seems to lack of some key questions (money willing to pay) and, anyway, it should be shown in the manuscript.

Also, paper needs a great revision regarding introduction and justification (but discussion it's ok for me).

Author Response

1.      I think it is an interesting paper, but lacks of a well-developed methodology. For example, the authors describe unnecessarily the well-known logistic regression model (a footnote or a reference would be enough), whilst variables (and their meaning) used in regressions are not explained. The survey sample is not enough big (but this could be logical if the geographical area is difficult to access -I don't know about this issue). Questionnaire seems to lack of some key questions (money willing to pay) and, anyway, it should be shown in the manuscript. Also, paper needs a great revision regarding introduction and justification (but discussion it's ok for me)

Response: Thanks for your constructive comments. Over all, the paper has been revised in response to the comment made. The methodology has been redeveloped in a more comprehensive way. The surveys took place in four villages (Ngalauka, Kalimanjila, Mpamantha and Nkhanga) as these villages are directly benefiaries of Chia lagoon natural ecosystems and also significantly affected by bad water quality at the lagoon. We had approximately 800 households. There were about 300 responses to the survey though some were incomplete with only 240 for CVM. I may agree with you that the sample size may be relatively smaller but statistically representative as compared to other studies available in literature. For example du Preez and Hosking [60] used a sample size of 96 respondents. Grillia et al [61] used the sample size of 134. Chae at al [62] collected sample of 161 survey questionnaires. Jiang and Rohendi [63] used sample size of 120 respondents randomly selected across the 9 districts. This has also been indicated in the manuscript. I have also added CV questionnaire with key questions on WTP amount. A section which estimate the WTP amount has also been added.

Reviewer 3 Report

This paper uses a various methods (surveys, focus groups, interviews and multi-variate logistic regression analysis) to assess factors and willingness-to-pay (WTP) for water quality improvement (WQI) for the Chia Lagoon in Malawi. While interesting and locally relevant, the paper suffers from various issues that need to be addressed before the paper can be considered for publication in Sustainability.

In particular: i) the ‘Introduction’ is thin, lacking a proper review of the international literature, no clear identification of gap(s) in knowledge and no clear objective definition; ii) the ‘Conceptual framework’ is also thin, lacking the theoretical principles underpinning and references to WTP as well as lacking the theoretical underpinnings and references of factors explaining WTP for water quality improvement; iii) the ‘Data collection and research methods’ should better describe, justify and reference the developed approach; iv) the ‘Results’ should be better presented and deeper described; and v) the ‘Conclusions’ should better describe novelties/innovations, relevance of the obtained results and corresponding policy implications of this study. In addition, the text needs to be reviewed (English) and edited (telegram style; use of paragraphs; sections).

The most critical aspect is, however, that it is totally unclear what the used 'WTP' is, given it is not expressed in monetary terms while the reference to WTP alludes to the fact it is. How is WTP measured in this study? What unit is used? What elicitation method has been applied? This fundamentally undermines the objective of this study, as it explicitly refers to WTP that is, generally, considered in monetary terms.

For the detailed comments, please refer to the attached annotated document (.pdf).

Author Response

1.      Title change recommended, e.g.:"Determinants and values of willingness to pay for water quality improvement: insights from Chia Lagoon, Malawi"- English needs to be thoroughly reviewed and text edited (telegram style; lack of paragraphs; sections)!

Thanks for your constructive comments. The suggestion has been taken and the title has been change from ‘Willingness to Pay for Improved Water Quality and Influencing factors: An Insight from Chia Lagoon, Malawi’ to “Determinants and values of willingness to pay for water quality improvement: insights from Chia Lagoon, Malawi"

2.      This paper uses a various methods (surveys, focus groups, interviews and multi-variate logistic regression analysis) to assess factors and willingness-to-pay (WTP) for water quality improvement (WQI) for the Chia Lagoon in Malawi. While interesting and locally relevant, the paper suffers from various issues that need to be addressed before the paper can be considered for publication in Sustainability. In particular: i) the ‘Introduction’ is thin, lacking a proper review of the international literature, no clear identification of gap(s) in knowledge and no clear objective definition; ii) the ‘Conceptual framework’ is also thin, lacking the theoretical principles underpinning and references to WTP as well as lacking the theoretical underpinnings and references of factors explaining WTP for water quality improvement; iii) the ‘Data collection and research methods’ should better describe, justify and reference the developed approach; iv) the ‘Results’ should be better presented and deeper described; and v) the ‘Conclusions’ should better describe novelties/innovations, relevance of the obtained results and corresponding policy implications of this study. In addition, the text needs to be reviewed (English) and edited (telegram style; use of paragraphs; sections).

Response: Thanks for your constructive comments. Over all, the paper has been revised in response to the comment made. Introduction has been restructured and well enriched with recent literature. Proper review of the international literature has been made and research gap has been clearly identified based on the knowledge obtained from literature. The last section of the introduction has demonstrated clear objective definition. The conceptual framework has been revised and theoretical principles underpinning and references to WTP has been clearly demonstrated. Theoretical underpinnings and references of factors explaining WTP for water quality improvement have been clearly explained. The conceptual framework has been enriched with recent international literatures. The methodology has been redeveloped in a more comprehensive way. I have also added CV questionnaire with key questions on WTP amount. A section which estimate the WTP amount has also been added. Section of the results has been revised and presented better. Description has been made supported with relevant literature. Conclusions has been revised and better described with novelties/innovations based on the results and policy implications. Finally, the manuscript has been reviewed by English native speaker for grammar, telegram style; use of paragraphs.

Round 2

Reviewer 1 Report

Dear Authors,

Many congratulations for your substantial effort, and improvement of your manuscript that now presents in a broader and more comprehensive manner the context factors, and conditions for willingness to pay for better water quality in Chia Lagoon. 

The addition of the Contingency Valuation method was a smart move, and it covers well many of my previous comments on the conceptual and analytical framework.

I wish you all the best success in your publication. It was a pleasure to review your improved work.

Best regards,

The Reviewer

Author Response

Many thanks for correction. It has improved the paper alot.

Reviewer 2 Report

I think my suggestions has been taken into account and now the paper has more scientific soundness.

Author Response

Thanks  for your comments.

Reviewer 3 Report

The paper has improved as compared to the original submission, and I would like to thank the authors for their efforts. Nevertheless, some serious methodological issues remain in the analysis that need serious attention and additional work. In particular:

- The authors have not convincingly shown what is novel/innovative in their study. I.e. key literature on WTP and water quality improvement has been overlooked.

- The Theoretical framework needs to be reinforced and justified based on this overlooked literature.

- Most importantly, the applied approach to estimate WTP and to assess factors influencing WTP is not conform established approaches that are abundant in this overlooked literature.

For the detailed comments, please refer to the attached annotated document (.pdf).

Author Response

Thanks for the comments. Please attached is the cover letter indicating all the issues that needed critical attention.

Round 3

Reviewer 3 Report

Serious methodological issues remain in the analysis that need serious attention and substantial additional work. In particular, the applied approach to estimate WTP and to assess factors influencing WTP is not conform established approaches that are abundant in literature. I.e. two separate analyses were performed, one being the WTP analysis and the other being the ‘factors influencing WTP’ analysis. It is well established to estimate WTP as a function of various variables that explain WTP (i.e. in one [1- or 2-stage] regression analysis; pooled or panel model; etc.) as to get an estimate on i) WTP, ii) factors influencing WTP, and iii) the extent to which these factors influence WTP.

Author Response

Thank you for your comment. your suggestions have been helpful. I have worked on the methodology to the best of my ability and I have carefully reviewed the methodology following various literature including those you recommended and I have made a tremendous change which has been reflected in yellow.

Thanks for you comments.